# CANT1 Is Involved in Collagen Fibrogenesis in Tendons by Regulating the Synthesis of Dermatan/Chondroitin Sulfate Attached to the Decorin Core Protein

**DOI:** 10.3390/ijms26062463

**Published:** 2025-03-10

**Authors:** Rina Yamashita, Saki Tsutsui, Shuji Mizumoto, Takafumi Watanabe, Noritaka Yamamoto, Kenta Nakano, Shuhei Yamada, Tadashi Okamura, Tatsuya Furuichi

**Affiliations:** 1Laboratory of Laboratory Animal Science and Medicine, Graduate School of Veterinary Sciences, Iwate University, Morioka 020-8550, Japan; a3124005@iwate-u.ac.jp; 2Laboratory of Laboratory Animal Science and Medicine, Co-Department of Veterinary Medicine, Faculty of Agriculture, Iwate University, Morioka 020-8550, Japan; shitanaga19@gmail.com; 3Department of Pathobiochemistry, Faculty of Pharmacy, Meijo University, Nagoya 468-8503, Japan; mizumoto@meijo-u.ac.jp (S.M.);; 4Laboratory of Veterinary Anatomy, School of Veterinary Medicine, Rakuno Gakuen University, Ebetsu 069-8501, Japan; t-watanabe@rakuno.ac.jp; 5Department of Mechanical Engineering, Ritsumeikan University, Kusatsu 525-8577, Japan; noritaka@se.ritsumei.ac.jp; 6Department of Laboratory Animal Medicine, Research Institute, National Center for Global Health and Medicine (NCGM), Shinjuku-ku 162-8655, Japanokamurat@ri.ncgm.go.jp (T.O.)

**Keywords:** CANT1, tendon, decorin, proteoglycan, glycosaminoglycan, dermatan sulfate, chondroitin sulfate, collagen fibril, Desbuquois dysplasia

## Abstract

Tendons are connective tissues that join muscles and bones and are rich in glycosaminoglycans (GAGs). Decorin is a proteoglycan with one dermatan sulfate (DS) or chondroitin sulfate (CS) chain (a type of GAG) attached to its core protein and is involved in regulating the assembly of collagen fibrils in the tendon extracellular matrix (ECM). Calcium-activated nucleotidase 1 (CANT1), a nucleotidase that hydrolyzes uridine diphosphate into uridine monophosphate and phosphate, plays an important role in GAG synthesis in cartilage. In the present study, we performed detailed histological and biochemical analyses of the tendons from *Cant1* knockout (*Cant1*^−/−^) mice. No abnormalities were observed in the tendons on postnatal day 1 (P1); however, remarkable hypoplasia was observed on P30 and P180. The collagen fibrils were more angular and larger in the *Cant1*^−/−^ tendons than in the control (Ctrl) tendons. In the *Cant1*^−/−^ tendons, the DS/CS content was significantly reduced, and the DC/CS chains attached to the decorin core protein became shorter than those in the Ctrl tendons. No abnormalities were observed in the proliferation and differentiation of tendon fibroblasts (tenocytes) in the *Cant1*^−/−^ mice. These results strongly suggest that CANT1 dysfunction causes defective DS/CS synthesis, followed by impairment of decorin function, which regulates collagen fibrogenesis in the tendon ECM. Multiple joint dislocations are a clinical feature of Desbuquois dysplasia type 1 caused by human *CANT1* mutations. The multiple joint dislocations associated with this genetic disorder may be attributed to tendon fragility resulting from CANT1 dysfunction.

## 1. Introduction

Tendons connect muscles to bones and transmit the mechanical force of muscle contraction to the skeleton, thereby allowing mobility and joint stability. Tendons have a unique collagenous extracellular matrix (ECM) with a hierarchical structure [1,2,3]. In this matrix, individual collagen molecules assemble into fibrils, and these fibrils assemble into fibers. These fibers, along with the tendon fibroblasts (tenocytes), are organized into fascicles that are bound together by connective tissue sheaths to form a tendon. This structure is crucial for facilitating force transmission and directly influences the mechanical functions of the tissue. Tenocytes synthesize specific ECM components, including collagen and proteoglycans (PGs). PGs consist of a core protein to which one or more glycosaminoglycan (GAG) chains are covalently attached. Many types of PGs, such as decorin, biglycan, fibromodulin, and lumican, are present in the tendon ECM [4,5,6]. These PGs, belonging to the small leucine-rich proteoglycan (SLRP) family, contain numerous leucine-rich repeats and are important regulators of ECM assembly and cell signaling in connective tissue.

GAGs, which are linear polysaccharides composed of repeating disaccharide units, are classified as chondroitin sulfate (CS), dermatan sulfate (DS), keratan sulfate (KS), heparan sulfate (HS), and hyaluronan (HA) [7,8]. GAGs, excluding HA, are present as side chains of PGs. CS is composed of repeating disaccharide units of glucuronic acid (GlcUA) and *N*-acetylgalacosamine (GalNAc), and DS is obtained by the epimerization of GlcUA in CS chains into iduronic acid (IdoUA), resulting in the formation of repeating units of IdoUA and GalNAc. HS and KS consist of the repeating disaccharide units of GlcUA/IdoUA and *N*-acetylglucosamine (GlcNAc) and of galactose and GlcNAc, respectively. HA, a free polysaccharide, is composed of repeating units of GlcUA and GlcNAc. In addition to their roles in ECM assembly, GAGs are critically involved in various biological functions, including cell adhesion and cellular signaling [9,10].

Most glycosylation reactions occur in the lumen of the endoplasmic reticulum (ER) and Golgi apparatus [11,12], in which glycosyltransferase utilizes nucleotide sugars as donor substrates and catalyzes the transfer of sugar moieties from a nucleotide sugar to a nucleophilic glycosyl acceptor molecule. Among all the nucleotide sugars, uridine diphosphate (UDP) sugars are the most important for this reaction. Most UDP sugars are synthesized in the cytosol and transported to the ER or Golgi apparatus by nucleotide sugar transporters (NSTs) [13,14,15]. NSTs carry UDP sugars coupled with the antiport of uridine monophosphate (UMP), a reaction product of luminal nucleoside diphosphatase that acts on UDPs produced in a glycosyltransferase reaction. The removal of UDP is also important for alleviating the feedback inhibition of glycosyl transferase [16,17]. Calcium-activated nucleotidase 1 (CANT1), a nucleoside diphosphatase present in the ER and Golgi, preferentially hydrolyzes UDP to UMP and phosphate [18,19,20]. Owing to its substrate preference and localization, CANT1 is considered one of the major luminal nucleoside diphosphatases involved in glycosylation.

Desbuquois dysplasia (DBQD) is a rare skeletal dysplasia inherited as an autosomal recessive trait; it is characterized by short limbs, a short stature, progressive scoliosis, a round face, and multiple joint dislocations [21,22,23,24]. DBQD is clinically heterogeneous and is classified into two types based on the presence (type 1) or absence (type 2) of the characteristic hand anomalies. These anomalies comprise an extra ossification center distal to the second metacarpal, delta phalanx, bifid distal thumb phalanx, and dislocation of the interphalangeal joints. An additional DBQD variant, called the DBQD Kim variant, has been proposed [25]. It is characterized by advanced carpal bone age and shortness of one or all metacarpal bones, with an elongated appearance of the phalanges but without an accessory ossification center. The causative genes for DBQD type 1 (DBQD1) and the Kim variant have been identified as *CANT1* [26,27,28], whereas those for DBQD type 2 have been identified as *XYLT1*, which encodes xylosyltransferase 1, an enzyme that functions as an initiator of GAG chain biosynthesis by transferring xylose from UDP-xylose to the core protein [29]. Indeed, impaired GAG synthesis has been observed in DBQD patients [28,29], as well as in both *Cant1* and *Xylt1* knockout (*Xylt1*^−/−^) mice [30,31,32,33]; it is therefore believed to be closely related to DBQD pathogenesis.

Multiple joint dislocations, which are occasionally caused by tendon rupture, are a clinical feature of DBQD. Therefore, we hypothesized that impaired GAG biosynthesis and subsequent SLRP dysfunction are responsible, at least in part, for tendon fragility and multiple joint dislocations in patients with DBQD1. To test this hypothesis, we performed detailed histological and biochemical analyses of the tendons of *Cant1*^−/−^ mice, a useful animal model of DBQD1 [30,31].

## 2. Results

### 2.1. Cant1^−/−^ Mice Exhibit Tendon Hypoplasia During Postnatal Development

Patellar tendon histology was examined using hematoxylin and eosin (HE) sections from mice on postnatal day 1 (P1), P30, and P180 (Figure 1A–C). There was no difference in tendon histology between the control (Ctrl) and *Cant1*^−/−^ mice on P1 (Figure 1A,D); however, the tendons were significantly thinner in the *Cant1*^−/−^ mice than in the Ctrl mice on P30 and P180 (Figure 1B–D). The cell density of the tendons was comparable between the Ctrl and *Cant1*^−/−^ mice on P1 and P180 (Figure 1E and Appendix A). However, the density of the *Cant1*^−/−^ tendons on P30 was significantly higher than that of the Ctrl tendons (Figure 1E and Appendix A).

The *Cant1*^−/−^ mice exhibited mildly wavy tails after infancy (Appendix A). The tail tendon on P180 was significantly thinner in the *Cant1*^−/−^ mice than in the Ctrl mice (Figure 2A,B). Fascicles of collagen fibers (collagen fascicles) are bound together by connective tissue sheaths to form the entire tendon (Figure 2C,D). The area and number of collagen fascicles in the *Cant1^−^*^/−^ tail tendons were significantly reduced as compared with those in the Ctrl tendons (Figure 2D,E). To examine the mechanical properties of the *Cant1*^−/−^ tendons, tensile tests were performed using collagen fascicles prepared from the tail tendons of the mice on P210. These experiments demonstrated a decrease in the maximum load in the *Cant1^−^*^/−^ tail tendons (Figure 2F), indicating functional depression. However, the tensile strength, strain at failure, and tangent modulus, which are measures of the elasticity of the material itself, indicating the load per unit area and deformation per unit length, were not significantly different between the Ctrl and *Cant1*^−/−^ mice (Figure 2F). These results suggest that the decrease in the maximum load in the *Cant1*^−/−^ tendons was due to the reduced tendon mass.

### 2.2. Fibrillogenesis Is Impaired in the Cant1^−/−^ Tendons

Electron microscopic analysis was performed using the transverse sections of the Achilles tendons from the mice on P180. The collagen fibrils were fairly uniform and circular in the wild-type tendons but were angular and cuboidal in the *Cant1*^−/−^ tendons (Figure 3A). The results of the image analysis showed that the area of the collagen fibril in the *Cant1*^−/−^ tendons was significantly increased compared to that in the Ctrl tendon (Figure 3B). The major diameter of the *Cant1*^−/−^ fibrils was also significantly larger than that of the Ctrl fibrils, but no significant difference was observed in the minor diameter between the two fibrils. The roundness of the *Cant1*^−/−^ fibrils was significantly reduced compared to that of the Ctrl fibrils. The *Cant1*^−/−^ fibrils appeared to be more tightly packed than the wild-type tendons (Figure 3A); indeed, the fibril density in the *Cant1*^−/−^ tendons was significantly higher than that in the Ctrl tendons (Figure 3C). When examining the distribution ratio of major collagen fibril diameters, the long fibrils were found to be increased, but the short fibrils were decreased in the *Cant1*^−/−^ tendons (Figure 3D).

### 2.3. DS/CS Content Is Markedly Reduced, and the Molecular Weight of Decorin Is Correspondingly Decreased in the Cant1^−/−^ Tendons

A disaccharide composition analysis was performed to determine the GAG content in the Achilles tendons of the mice on P330 (Table 1 and Appendix A). The CS and DS were more abundant in the tendons, and their contents in the *Cant1*^−/−^ tendons were significantly reduced compared to those in the Ctrl tendon. This decrease was more pronounced in the DS content than in the CS contents. The sulfation patterns in the CS and DS were comparable between the Ctrl and *Cant1*^−/−^ mice (Appendix A). The HA content was less than 1/10th of the CS/DS content in the Ctrl tendons. The HA content was significantly lower in the *Cant1*^−/−^ tendons than in the Ctrl tendons. The HS and KS contents were below the detection limits.

Decorin is an SLRP with a DS/CS chain attached to its core protein. Decorin in the tail tendons of the mice on P30 and P180 was detected using Western blot analysis (Figure 4A). The molecular weight of decorin detected on both P30 and P180 was clearly lower in the *Cant1*^−/−^ tendon extracts than in the Ctrl tendon extracts; however, the molecular weight of the core protein detected in the chondroitinase ABC-treated extracts did not differ between the two tendon extracts (Figure 4A). Therefore, the DS/CS chain attached to the decorin core protein produced in the *Cant1*^−/−^ tendons was shorter than that in the Ctrl tendons. In both the Ctrl and *Cant1*^−/−^ tendons, the molecular weight of decorin in the tendon extracts was higher on P30 than on P180 (Figure 4B).

### 2.4. Proliferation and Differentiation Are Not Affected in the Cant1^−/−^ Tendon Cells

A BrdU incorporation study was conducted to examine the proliferative activity of tenocyte in the patellar tendons of the mice on P4 and P10 (Figure 5A). The activity was comparable between the Ctrl and *Cant1*^−/−^ mice on both P4 and P10. Finally, the expression levels of the tendon marker genes *Scx*, *Mkx*, *Col1a1*, *Tnmd*, *Dcn*, and *Fmod* were determined using real-time PCR for the tail tendons on P40 (Figure 5B). The expression levels of all marker genes were also comparable between the Ctrl and *Cant1*^−/−^ mice. Therefore, proliferation and differentiation in the tenocytes were not affected in the *Cant1*^−/−^ mice.

## 3. Discussion

In the latest nosology of genetic skeletal disorders, 771 diseases are registered across 41 groups, and DBQD1 caused by *CANT1* mutations is classified in Group 5, “Dysplasia with multiple joint dislocations” [34]. This group includes diseases caused by mutations in genes encoding glycosyltransferases involved in the synthesis of tetrasaccharides that bridge the GAG chains and proteoglycan core proteins (*XYLT1*, *B3GALT6*, *B4GALT7*, and *B3GAT3*) and in the synthesis of CS chains (*CSGALNACT1*). These findings indicate that GAG abnormalities are a risk factor for developing joint dislocations. CANT1 has also been reported to be involved in GAG synthesis in cartilage [28,30,31]. Therefore, we hypothesized that the GAG abnormalities in the tendons of DBQD1 patients cause fragility and predispose them to joint dislocations. As expected, the tendons from the *Cant1*^−/−^ mice, an animal model of DBQD1, showed abnormalities in fibrogenesis, along with angular and larger collagen fibrils, a decreased DS/CS content, and shortening of the DC/CS chains attached to the decorin core protein.

In the *Cant1*^−/−^ tendons, the DS/CS content was significantly reduced (DS reduction was more pronounced), and the DS/CS chains attached to the decorin core protein were shortened compared to those in the Ctrl tendons. CANT1 preferentially catalyzes the conversion of UDP to UMP in the ER/Golgi lumen [26,28]. The generated UMPs are used as substrates for the antiport mediated by the NSTs [13,14,15]. Therefore, the reduced UMP production due to CANT1 deficiency should reduce the NST activity. As a result, there are limited nucleotide sugars in the ER/Golgi lumen, leading to a reduced DS/CS content. However, previous and present studies have shown that CANT1 deficiency does not completely suppress the GAG synthesis in cartilage and tendons [30,31]. Therefore, it is possible that the other nucleotidases complement the CANT1 functions or that other mechanisms supply UDP sugars to the ER/Golgi lumen.

Approximately 80% of PGs present in the tendons are decorins [35]. A single DS or CS chain is attached to the core protein, and the DS-bound-type decorin is the most common type found in tendons. The tendons from mice deficient in *Dcn* encoding the decorin core protein exhibit abnormalities, such as the co-existence of larger and smaller collagen fibrils and increased fusion of adjacent fibrils [36,37]. Collagen fibrils are formed by the covalent cross-linking of short collagen molecules called protofibrils. As the collagen fibrils enter the maturation stage, their diameter increases, and the fusion between them occurs [2,3,35]. Given that decorin binds to protofibrils [38], it is thought to regulate the steps involved in the construction of collagen fibrils from protofibrils. Furthermore, decorin inhibits fibril maturation and fibril-to-fibril fusion, thereby increasing the diameter of *Dcn*^−/−^ fibrils [9,37,39]. The increased diameter of the *Cant1*^−/−^ fibrils may be explained, at least in part, by the dysregulated decorin function caused by the shortening of the DS/CS chain. The tendons contain other SLRPs, including biglycan, lumican, and fibromodulin, which are also involved in tendon fibrogenesis [4,5,6], suggesting that CANT1 deficiency also affects the GAG side chains of these SLRPs. The defective functions of these SLRPs may affect the pathogenesis of *Cant1*^−/−^ tendons.

During DS formation, GlcUA in the CS precursor is converted to IdoUA by epimerization to form IdoUA and GalNAc repeating units. This conversion is mediated by DS epimerases encoded by *DSE* and *DSE1* [40,41]. The CS/DS, CS, and DS contents in the Achilles tendons of the *Cant1*^−/−^ mice were reduced to 31%, 56%, and 13%, respectively, of those of the Ctrl mice, suggesting that the proportion of CS converted to DS was reduced. In mammalian cells, UDP inhibits glycosyltransferase activity, whereas UMP does not [16,17]. Therefore, accumulated UDPs or unknown defects in the Golgi lumen of *Cant1*^−/−^ cells may also inhibit the activity of DS epimerase, leading to a reduction in the proportion of CS converted to DS. Dermatan chains are matured by sulfation reactions catalyzed by dermatan 4-*O*-sulfotransferase-1 (D4ST1) and uronosyl 2-*O*-sulfotransferase (UST), which are encoded by *CHST14* and *UST*, respectively. The musculocontractural Ehlers–Danlos syndrome (mcEDS), which is characterized by kyphoscoliosis, muscular hypotonia, skin hyperextensibility and fragility, joint hypermobility, and multiple joint dislocations, is caused by *DSE* or *CHST14* mutations [42,43,44]. The *Dse*^−/−^ mice were smaller, with a 20–30% reduction in body weight at birth as compared to the wild-type mice; they also had wavy tails until P30 [45]. The diameters of the collagen fibrils in the skin and tail tendons of the *Dse*^−/−^ mice were higher than those in the wild-type mice. Wavy tails were also observed in the *Chst14*^−/−^ mice. These findings suggest that the DS chain abnormalities resulting from *DSE* or *DHST14* mutations, as well as *CANT1* mutations, may lead to defective fibrogenesis and some degree of fragility in tendons.

In the skin of patients with mcEDS caused by *CHST14* mutations (mcEDS-*CHST14*), the decreased enzymatic activity of dermatan sulfation causes the conversion of DS to CS [43]. Thus, most of the decorin molecules produced in the patients were converted from a DS-bound form to a CS-bound form. Electron microscopic analysis of normal skin showed that decorin is bound to curved DS chains, which are in close contact with adherent collagen fibrils, and that fibrils are tightly packed [46,47]. Contrarily, in mcEDS-*CHST14* patients and *Chst14*^−/−^ mice, the CS chains bound to decorin were deformed into a linear shape and protruded from the collagen fibrils. Meanwhile, the collagen fibrils were disjointed, thereby increasing the distance between the neighboring fibrils. This conformational change in the decorin side chain is thought to be responsible for the spatial disorganization of the collagen assembly and skin fragility caused by *CHST14* mutations. In the *Cant1*^−/−^ tendons, the distance between the neighboring fibrils did not increase but it appeared to decrease. The *Cant1*^−/−^ fibrils became angular and larger, whereas the size and shape of the *Chst14*^−/−^ fibrils appeared normal. Therefore, DS/CS chain shortening due to CANT1 dysfunction and DS-to-CS chain conversion due to D4ST1 dysfunction appear to have different effects on fibrogenesis. DS/CS chain shortening may make it impossible to maintain an appropriate distance between fibrils, and the close distance between the fibrils may be partially attributed to their angular shape.

The role of decorin in the mechanical properties of mature tendons remains controversial, and mechanical tests using *Dcn*^−/−^ tendons have yielded mixed results. The strength and stiffness in the flexor digitorum longus tendon of the *Dcn*^−/−^ mice on P60 were similar to those of the wild-type mice, but they were significantly reduced on P150 [36]. The mechanical properties of the tail and flexor digitorum longus tendons of the *Dcn*^−/−^ mice on P240–300 were comparable to those of the wild-type mice; however, in the patellar tendon, the modulus (tangent modulus) was increased [48]. The role of SLRP in the mechanical properties of tendons may vary in complex ways, depending on the tendon location, mouse age, and other factors. In the tensile tests performed in the present study, the maximum load on the *Cant1*^−/−^ tail tendons decreased. However, this was a result of the reduced mass of the *Cant1*^−/−^ tail tendons and does not reflect the qualitative fragility of the tail tendons. Detailed studies using various types of tendons and mice of different ages are needed to clarify the role of CANT1 in the mechanical strength of tendons.

PGs not only play a role in ECM assembly but also regulate cell signaling through various growth factors [9,10]. Decorin regulates tumor growth by binding to TGF-β, which inhibits its activity, and to the EGF receptor, which acts as an antagonist [49,50,51]. Although both growth factors are involved in tendon development and healing processes, no effects on proliferation or differentiation were observed in the tenocyte of the *Cant1*^−/−^ tendons. Both growth factors bind to the decorin core protein [49,50,51]; therefore, the shortening of the DS side chain did not seem to have a considerable effect on the modulatory function of decorin on the growth factor activity. Meanwhile, HS exerts a considerable effect in the control of cell proliferation. Given that the HS content is extremely low in tendons, it may have no effect on the proliferation of *Cant1*^−/−^ tenocytes [52]. In the newborn stage, *Cant1*^−/−^ mice were slightly smaller than the wild-type mice, and no histological abnormalities were observed in the tendons. During the early postnatal development, dwarfism and tendon hypoplasia, including a decrease in the area and number of collagen fascicles, became prominent in the mutants. Given that there were no abnormalities in the proliferation and differentiation of *Cant1*^−/−^ tenocytes, the tendon hypoplasia in the *Cant1*^−/−^ mice was most likely a phenotype secondary to skeletal dwarfism. The main reason for the increased cell density in the *Cant1*^−/−^ tendons on P30 appears to be the reduced ECM space. CS provides water retention and elasticity to tissues, leading to increased ECM space [53,54], suggesting that a decreased CS content in *Cant1*^−/−^ tendons leads to decreased ECM space.

CANT1 may be involved in glycan synthesis beyond GAGs and has also been reported to be involved in protein quality control and folding in neuroblastoma cell lines [55]. Furthermore, a soluble secreted form is present in CANT1 [18,19], whereas uridine nucleotides and UDP sugars function as endogenous ligands for the P2Y receptor [56]. Thus, soluble secretory CANT1 may modulate cellular responses to extracellular UDP. CANT1 is also overexpressed in many tumors and regulates tumor cell proliferation, attracting attention as a potential prognostic marker [57,58]. There may be other roles for CANT1 in tendons, apart from GAG synthesis.

Recently, considerable progress has been made in the development of therapies for skeletal dysplasia, including enzyme replacement, cellular therapy, gene therapy, and pharmacological therapies [59,60]. We present the possibility that DS replacement in the tendons may be effective in improving the quality of life of DBQD1 patients. *Cant1*^−/−^ mice are an appropriate animal model for developing this therapy.

## 4. Materials and Methods

### 4.1. Animals

The production of the *Cant1*^−/−^ mice used in this study has previously been described [30]. These mice were generated by crossing *Cant1*^+/−^ breeding pairs. Given that the *Cant1*^+/−^ mice did not present an overt phenotype, the wild-type and *Cant1*^+/−^ littermates were used as Ctrl mice. The mice were housed in a temperature-controlled room with a 12 h/12 h light/dark cycle and fed standard mouse laboratory chow with free access to water. The mice were sacrificed with an overdose of pentobarbital or decapitation. The correspondence between the types of analyses, types of tendon used, and mouse age is shown in Appendix A. The rationality behind the mouse ages used in the respective analyses is described in the Appendix A.

### 4.2. Macroscopic and Histological Analyses

After skin removal, the respective tendons were observed macroscopically. The hind limbs, including the Achilles tendon, patellar tendon, and tail, were fixed in a 4% paraformaldehyde solution in phosphate-buffered saline (PBS), decalcified in 10% EDTA for 1 week at 4 °C, and embedded in paraffin. HE staining was performed using 3 μm paraffin sections, according to standard protocols. For the cell proliferation analysis, the BrdU solutions were intraperitoneally injected at 100 μg of BrdU/g body weight into the intraperitoneal space of the mice on P1, P2, and P3 or P8 and P9. The hind limbs were collected on P4 or P10, and paraffin sections were prepared. A BrdU Immunohistochemistry Kit (Abcam, Cambridge, UK) was used according to the manufacturer’s protocol to detect BrdU incorporation into the cells.

### 4.3. Transmission Electron Microscopy (TEM)

Fresh Achilles tendons were prefixed overnight in 3.0% glutaraldehyde in 0.1 M phosphate buffer (pH 7.4) at room temperature. After washing with 0.1 M phosphate buffer, the samples were postfixed in 1.0% osmium tetroxide in 0.1 M phosphate buffer for 1 h at room temperature. After washing with distilled water, the samples were dehydrated in graded ethanol and transferred to QY-1 (Nisshin EM, Tokyo, Japan). The samples were embedded in Quetol 812 (Nisshin EM) and cut into ultrathin sections using an ultramicrotome (EMUC7; Leica, Wetzlar, Germany) equipped with a diamond knife. Ultrathin sections (~80 nm thick) were mounted on a copper grid and consecutively stained with 0.2% tannic acid in 0.1 M phosphate buffer for 15 min, 1.0% uranyl acetate for 10 min, and 1.0% lead citrate for 3 min. Between each staining step, the ultrathin sections were washed with distilled water. A transmission electron microscope (JEM-1220; JEOL, Tokyo, Japan) operating at an accelerating voltage of 80 kV was used for this experiment. We measured the diameter and circularity of approximately 1200 collagen fibrils from four TEM images of two mice in each group using ImageJ (version 1.48v; National Institutes of Health, Bethesda, MD, USA).

### 4.4. Tensile Testing

The fascicles were resected from the tail tendons. The fascicular diameters were measured using a video microscope (TG30TV-2, Shodensha, Osaka, Japan). The cross-sectional area was determined from the diameter, assuming that the cross-section was circular. For mechanical testing, a specially designed microtensile tester was mounted on a video microscope. Both ends of each fascicle were clamped using an acrylic grip. One of the grips was fixed to a load cell (LTS-500GA, Kyowa Electronic Instruments, Tokyo, Japan), whereas the other was fixed to the crosshead of the linear actuator (CM420-3C, Oriental motor, Tokyo, Japan). The load cell had a sensitivity of 0.005 N in the measurement range of 0–5 N. The gauge length of the fascicles was determined as the distance between the grips. The fascicular gauge lengths were almost constant at 2.65 ± 0.23 mm (mean ± SD). The fascicles and grips were immersed in a physiological saline solution. The fascicle was stretched until failure by moving the actuator at a speed of 0.1 mm/s. The stress was calculated by dividing the applied load by the fascicle’s cross-sectional area. The strain was determined by dividing the deformation by the initial fascicle length. From these data, the stress–strain relationship was obtained. The tensile strength and strain at failure were defined as the stress and strain at the fascicle’s failure point, respectively. The tangent modulus was defined as the slope of the linear portion of the stress–strain curve. The maximum load was calculated by multiplying the tensile strength of the fascicle by the tendon’s cross-sectional area, which was measured from the HE-stained sections of the tail.

### 4.5. Disaccharide Composition Analysis of the GAGs

The total disaccharide content of CS/DS, CS, DS, HS, KS, and HA in the tail tendons of the mice on P330 was determined as described previously [8]. In brief, the tendon samples were homogenized, sonicated, and exhaustively treated with activase E (Kaken Pharm., Kyoto, Japan) to degrade the proteins. The total amount of protein in the sonicated samples was determined using a Micro BCA Protein Assay Kit (Thermo Fisher Scientific, Waltham, MA, USA). The remaining proteins and peptides were precipitated using trichloroacetic acid followed by extraction with ether to remove trichloroacetic acid. The resultant crude GAG–peptide fractions were desalted using an Amicon Ultra-0.5 3 K unit (Millipore, Billerica, MA, USA) and treated individually with a mixture of chondroitinase ABC and AC-II (Seikagaku Corp., Tokyo, Japan); chondroitinase AC-I and AC-II (Seikagaku Corp.); heparinase I (IBEX Pharmaceuticals, Montreal, Canada), heparinase II (R&D Systems, Minneapolis, MN, USA) and heparitinase I (Seikagaku Corp.); chondroitinase B (Seikagaku Corp.); and keratanase II (Seikagaku Corp.) for the analysis of the disaccharide composition of CS/DS, CS/HA, HS, DS, and KS, respectively. The digests were labeled with a fluorophore 2-aminobenzamide (2AB), and aliquots of the 2AB-derivatives of GAG disaccharides were analyzed using anion-exchange HPLC on a PA-G column (YMC Co., Kyoto, Japan). The unsaturated disaccharides detected in the digests were identified by comparing them with the elution positions of the authentic 2AB-labeled disaccharide standards.

### 4.6. Western Blot Analysis

The tail tendons were cut into small pieces, and proteins were extracted using an extraction buffer consisting of 4 M guanidine–HCl and 50 mM sodium acetate (pH 5.8) with a proteinase inhibitor (Sigma-Aldrich, St. Louis, MO, USA) at 4 °C for 48 h with shaking. The supernatant was passed through 7 K MWCO Zeba Spin Desalting Columns (Thermo Fisher Scientific, Waltham, MA, USA) and changed to a digestion buffer consisting of 150 mM Tris-HCl and 150 mM NaCl (pH 7.3). The samples were digested with 0 or 50 mU chondroitinase ABC (chABC, Sigma-Aldrich) for 24 h at 37 °C. Then, the samples (chABC-nontreated extract; 5 μg protein per lane, chABC-treated extract; 1 μg protein per lane) were separated on 12% Mini-PROTEAN TGX Gels (Bio-Rad, Hercules, CA, USA) and transferred to PVDF membranes. The membranes were incubated with 5% BSA in TBS-T to block nonspecific binding. The membrane was then incubated with rabbit anti-decorin antibody (EPR24097-105, Abcam) at a 1:1000 dilution with Can Get Signal Solution 1 (TOYOBO, Tokyo, Japan) at 4 °C for 16 h, followed by HRP-conjugated goat anti-rabbit IgG (Proteintech, Rosemont, IL, USA) at a 1:10,000 dilution with Can Get Signal Solution 2 at room temperature for 1 h. The development was performed with the Clarity TM Western ECL Substrate (Bio-Rad) and imaged using the ChemiDoc TM MP System (Bio-Rad).

### 4.7. Real-Time PCR

Tendon samples frozen in liquid nitrogen were homogenized using T10 basic ULTRA-TURRAX^®^ (IKA, Staufen, Germany). Total RNA was extracted using RNAiso Plus (Takara Bio, Shiga, Japan). Equal amounts of total RNA were reverse-transcribed into cDNA using the ReverTra Ace qPCR RT Master Mix (TOYOBO). Each reverse transcription reaction (1 μL) was used as a template for real-time PCR using the THUNDERBIRD SYBR qPCR Mix (TOYOBO). SYBR Green PCR and real-time fluorescence detection were performed using the StepOnePlus^TM^ Real-Time PCR System (Thermo Fisher Scientific). The primer sequences used are listed in Appendix A.

### 4.8. Statistical Analyses

Student’s *t*-test was used to determine the significance of differences between the Ctrl and *Cant1*^−/−^ mice. Statistical significance was defined as *p* < 0.05.

## 5. Conclusions

The *Cant1*^−/−^ mice exhibited tendon hypoplasia during postnatal development. The *Cant1*^−/−^ tendons showed abnormalities in fibrogenesis, along with angular and larger collagen fibrils, decreased DS/CS content, and shortening of DC/CS chains attached to the decorin core protein. These findings indicate that CANT1 is critical for collagen fibrogenesis in tendons. Multiple joint dislocations in patients with DBQD1, a genetic disorder caused by *CANT1* mutations, may be attributed to tendon fragility resulting from CANT1 dysfunction.

## Figures and Tables

**Figure 1 ijms-26-02463-f001:**
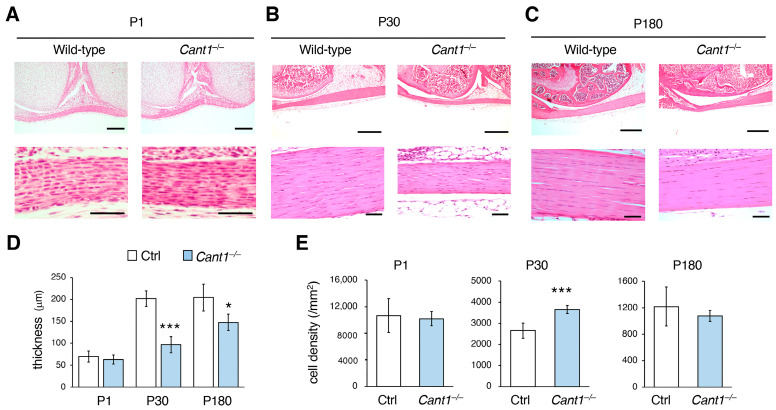
Postnatal tendon development is impaired in *Cant1*^−/−^ mice. Hematoxylin and eosin (HE)-stained sections of the patellar tendons from the Ctrl and *Cant1*^−/−^ mice on postnatal day1 (P1) (**A**), P30 (**B**), and P180 (**C**). Scale bar = 50 μm ((**A**–**C**): lower panels), 200 μm ((**A**): upper panels), and 500 μm ((**B**,**C**): upper panels). (**D**) Quantification of patellar tendon thickness in the Ctrl and *Cant1*^−/−^ mice. (**E**) Quantification of cell density in the patellar tendons of the Ctrl and *Cant1*^−/−^ mice. Cell density was measured using HE-stained sections of the patellar tendon. Values represent means ± SD (*n* = 4–5). * *p* < 0.05 and *** *p* < 0.001 between the Ctrl and *Cant1*^−/−^ mice.

**Figure 2 ijms-26-02463-f002:**
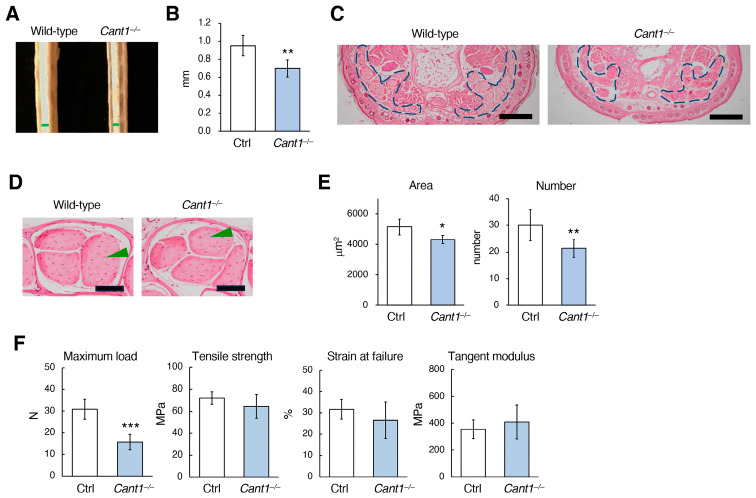
Histological and mechanical analyses of the tail tendons. The tails on postnatal day 180 (P180) and P210 were used for histological (**A**–**E**) and biomechanical (**F**) analyses, respectively. (**A**) Appearance of skinned tails in the wild-type and *Cant1*^−/−^ mice. Green bars indicate the tendon thickness. (**B**) Quantification of tail tendon thickness in the Ctrl and *Cant1*^−/−^ mice. Values represent the means ± SD (*n* = 5). (**C**) HE-stained sections of the ventral aspect of the tail from the wild-type and *Cant1*^−/−^ mice. The area surrounded by the blue dotted line represents the entire tendon, which is a major group of collagen fascicles. Scale bar = 500 μm. (**D**) HE-stained sections of the tail tendon from the wild-type and *Cant1*^−/−^ mice. Green arrowheads indicate collagen fascicles. Scale bar = 50 μm. (**E**) Quantification of the area and number of collagen fascicles in the tail tendons of the Ctrl and *Cant1*^−/−^ mice. Values represent means ± SD (area, *n* = 5; number, *n* = 6). We calculated the mean value of eight areas in the pictures for one mouse and compared them for the Ctrl and *Cant1*^−/−^ mice. (**F**) Quantification of the maximum load, tensile strength, strain at failure, and tangent modulus of the tail tendons from the Ctrl and *Cant1*^−/−^ mice. Values represent means ± SD (*n* = 5). * *p* < 0.05, ** *p* < 0.01, and *** *p* < 0.001 between the Ctrl and *Cant1*^−/−^ mice.

**Figure 3 ijms-26-02463-f003:**
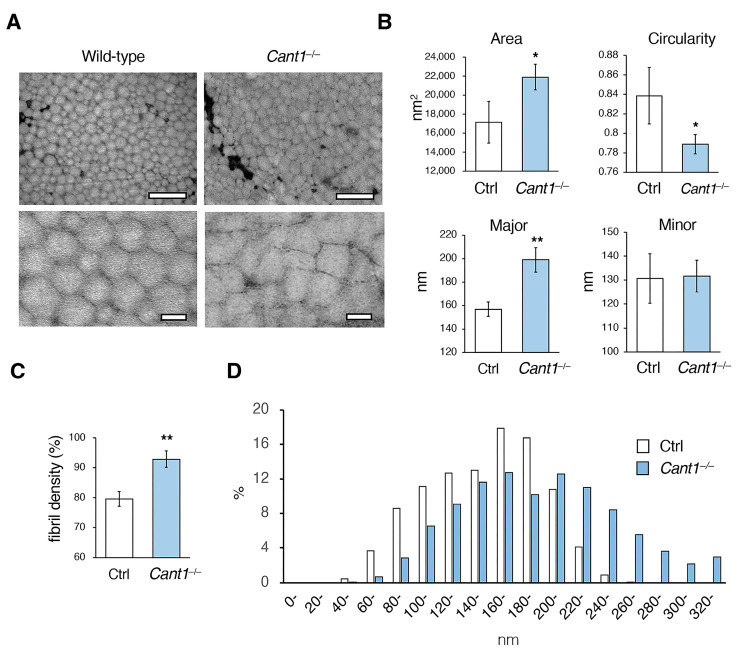
Collagen fibrillogenesis is impaired in the tendons of the *Cant1*^−/−^ mice. (**A**) Transverse section transmission electron microscopy (TEM) images of the Achilles tendons from the wild-type and *Cant1*^−/−^ mice on postnatal day 180 (P180). The upper and lower panels show the low- and high-magnification images, respectively. Scale bar = 500 nm (upper panel) and 100 nm (lower panel). (**B**) Quantification of the area, circularity, and major and minor diameters of the collagen fibrils in the TEM images. A circularity value of 1.0 indicates a perfect circle. Values represent means ± SD (*n* = 3). We calculated the mean values of collagen fibrils (approximately 500) in the images for one mouse and compared them for the Ctrl and *Cant1*^−/−^ mice. (**C**) Quantification of the fibril density in TEM images. Values represent means ± SD (*n* = 3). (**D**) Distribution of the collagen fibril major diameter in the Achilles tendons. * *p* < 0.05 and ** *p* < 0.01 between the Ctrl and *Cant1*^−/−^ mice.

**Figure 4 ijms-26-02463-f004:**
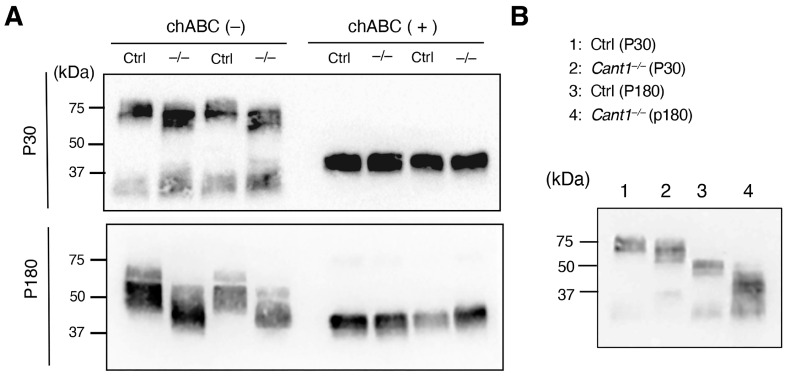
The molecular weight of decorin decreased in the tendons of the *Cant1*^−/−^ mice. (**A**) Western blot images of decorin in the tail tendon extracts from the Ctrl and *Cant1*^−/−^ mice on postnatal day 30 (P30) and P180. Ctrl, control mice; −/−, *Cant1*^−/−^ mice; chABC (−), non-chondroitinase ABC-treated extract; chABC (+), chondroitinase ABC-treated extract. (**B**) Comparison of the molecular weights of decorin in the tail tendon extracts from the mice on P30 and P180.

**Figure 5 ijms-26-02463-f005:**
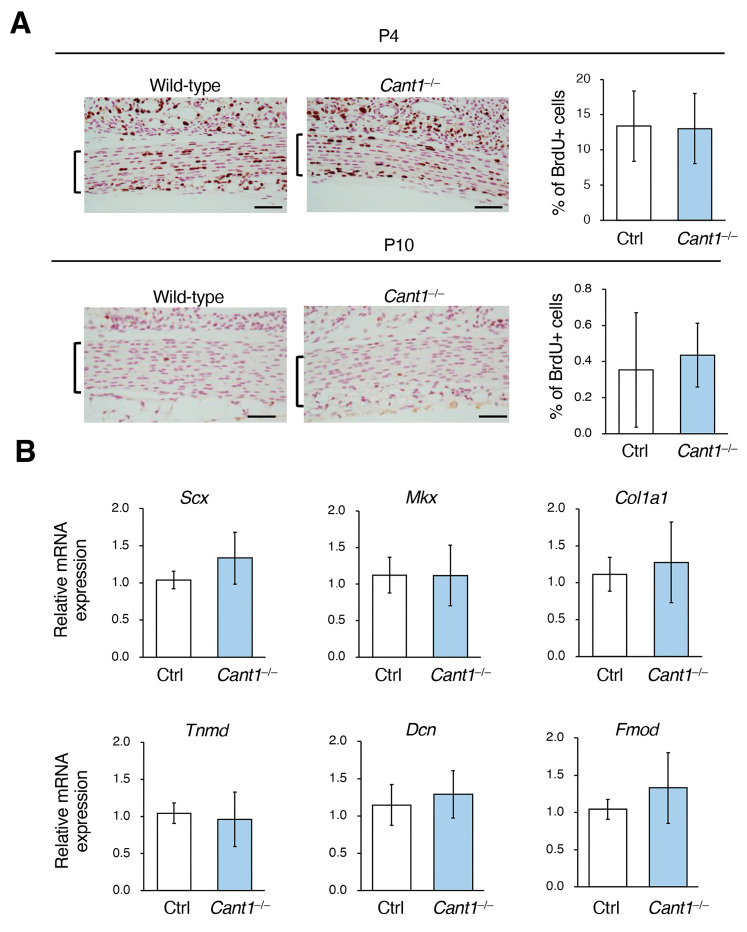
Proliferation and differentiation in the tenocytes of the *Cant1*^−/−^ mice were not affected. (**A**) BrdU staining of the sections including the patellar tendons from the BrdU-treated mice on P4 and P10, and graphs showing the frequency of BrdU-positive cells in the patellar tendons. The brackets indicate the patellar tendon thickness. Values represent means ± SD (*n* = 4). Scale bar = 50 μm. (**B**) The mRNA expression levels of the tendon marker genes (*Scx*, *Mkx*, *Col1a1*, *Tnmd*, *Dcn*, and *Fmod*) in the Achilles tendons from the Ctrl and *Cant1*^−/−^ mice on postnatal day 40 (P40) were measured using quantitative real-time PCR. Relative quantification was performed using the ΔΔ CT method, with *Gapdh* used as a reference gene. Values represent means ± SD *(n* = 6).

**Table 1 ijms-26-02463-t001:** The GAG levels in the Achilles tendon from the Ctrl and *Cant1*^−/−^ mice on P330.

GAG type	GAG level (pmol disaccharide/μg protein)		*Cant1*^−/−^/Ctrl ratio	*p*-Value
	Ctrl	*Cant1* ^−/−^		
CS/DS	194 ± 32.7	59.4 ± 7.0	0.31	0.00020
CS	84.4 ± 26.0	47.4 ± 13.8	0.56	0.045
DS	94.2 ± 17.2	11.9 ± 6.7	0.13	0.00011
HS	n.d.	n.d.	-	-
KS	n.d.	n.d.	-	-
HA	13.4 ± 4.1	7.84 ± 1.7	0.59	0.048

GAG contents were calculated based on the peak area in chromatograms of digests with a mixture of chondroitinase ABC and AC-II (CS/DS), chondroitinase AC-I and AC-II (CS/HA), chondroitinase B (DS), heparinase I, II and heparitinase I (HS), and keratanase II (KS). Abbreviations: GAG, glycosaminoglycan; CS, chondroitin sulfate; DS, dermatan sulfate; HS, heparan sulfate; KS, keratan sulfate; HA, hyaluronan; n.d., not detected.

## Data Availability

The data presented in this study are available upon request from the corresponding author.

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
