# Peer review of "CANT1 Is Involved in Collagen Fibrogenesis in Tendons by Regulating the Synthesis of Dermatan/Chondroitin Sulfate Attached to the Decorin Core Protein"

_ijms, 2025, doi:10.3390/ijms26062463_

Round 1
Reviewer 1 Report
Comments and Suggestions for Authors
The authors present a very interesting study on the role of glycosaminoglycans in tendon. They used an established mice model with Cant1 knockout and performed histological, immunohistochemical, electronmicroscopical, biochemical as well as biomechanical tests of mice tendons. This is a follow up on the FEBS open study on cartilage (Ref. 30). The manuscript is well written and the study design, the used methods and the results are clearly described. The figures need some improvement.
There are some points that need to be addressed:
- Abstract: Please add joint to the last sentence: „ multiple dislocations ”
- 1, 2,3, 5: Histologies and TEM look out of focus/blurry. Please check.
- Please add a quantification of tendon thickness.
- Please change all graphs: a presentation of the individual values will be better, dot/scatter plots
- 2E: “Eight areas and two numbers were counted in the sections from each mouse.” Please use the mean value for each mouse tendon and not the individual values of all areas and numbers.
- 2 Fibrillogenesis is impaired in the Cant1−/− tendons. “The long diameter…“ and “..the long fibrils were found to be increased, but the short fibrils were decreased ..”. Please correct. “long” seems to be wrong in this context. Use major and minor as in the graph.
- “Decorin is an SLRP with a DS/DS chain..” Should it read …DS/CS chain?
- Discussion. “Contrarily, CANT1 has been reported..” Why contrarily?
- The authors often use “contrarily” in the discussion and it seems not always correct. Please check.
- “The role of SLRP in the mechanical properties of tendons may vary in complex ways, depending on the tendon location, mouse age, and other factors.” I agree that location and age might influence the tendon properties. Therefore, the authors must discuss the possible effect of age in more detail, especially as they used different age groups for the different analysis. What was the rational for using different ages for the different methods? Please add a table to the Material and method section showing the method and the used tendon and age.
- 3 Material and Methods. “…for 2 h at room temperature overnight.” 2 hours or overnight, not both.
- “..1,200 collagen fibrils from four TEM images of two mice..” Was the mean value of each mice used for the statistics? N=2? If not, a t-test is not appropriate.
- 4. Tensile testing. Please provide more information on the test apparatus. Which measurement range/sensitivity had the load cell?
- Please provide the ARRIVE documentation.
Author Response
Comment 1: Abstract: Please add joint to the last sentence: „ multiple dislocations ”
Response: We have corrected the main text, as instructed.
Comment 2: 1, 2,3, 5: Histologies and TEM look out of focus/blurry. Please check.
Response: We have submitted the high-quality figures to the Editorial Office.
Comment 3: Please add a quantification of tendon thickness.
Response: We quantified the tendon thickness and added the corresponding graph (Figure 1D). Accordingly, the graphs of cell density in the Achilles tendons have been moved to Supplemental Information (Figure S1).
Comment 4: Please change all graphs: a presentation of the individual values will be better, dot/scatter plots.
Response: We agree that graphs with individual values or dot/scatter plots are preferable. However, because we used Excel to create the graphs, we were unable to create them. As many papers published in Int. J. Mol. Sci. present bar graphs without individual values, we would like to present the graphs as they are.
Comment 5: 2E: “Eight areas and two numbers were counted in the sections from each mouse.” Please use the mean value for each mouse tendon and not the individual values of all areas and numbers.
Response: We have corrected as follows: “Values represent means ± SD (area, n = 5, number, n = 6). We calculated the mean value of eight areas in the pictures for one mouse and compared them for the Ctrl and Cant1–/– mice.” Fig. 2E and the corresponding legend have been revised accordingly.
Comment 6: 6.2 Fibrillogenesis is impaired in the Cant1−/− tendons. “The long diameter…“ and “..the long fibrils were found to be increased, but the short fibrils were decreased ..”. Please correct. “long” seems to be wrong in this context. Use major and minor as in the graph.
Response: We have corrected the main text, as instructed.
Comment 7: “Decorin is an SLRP with a DS/DS chain..” Should it read …DS/CS chain?
Response: We have corrected the DS/DS chain to DS/CS chain.
Comment 8: Discussion. “Contrarily, CANT1 has been reported..” Why contrarily?
Comment 9: The authors often use “contrarily” in the discussion and it seems not always correct. Please check.
Response: We have corrected each “contrarily” to the proper conjunctions.
Comment 10: “The role of SLRP in the mechanical properties of tendons may vary in complex ways, depending on the tendon location, mouse age, and other factors.” I agree that location and age might influence the tendon properties. Therefore, the authors must discuss the possible effect of age in more detail, especially as they used different age groups for the different analysis. What was the rational for using different ages for the different methods? Please add a table to the Material and method section showing the method and the used tendon and age.
Response: Following the instructions of Reviewer 1, the correspondence between the types of analyses, types of tendon used, and mouse age is shown in Table S5. Further, the rationality behind the mouse ages used in the respective analyses is described in Supplemental Information (Information S1). However, it remains unclear which type of tendon, location, and the mouse age are appropriate for each analysis. We are also aware that it is desirable to perform all analyses at the same and at several mouse ages. However, this was not possible because of factors such as the effects of the COVID-19 pandemic, available space in laboratory animal facilities, and effective use of a single mouse. The detailed effects of tendon type, location, and mouse age on the pathology of Cant1−/− tendons will be the subject of future studies.
Comment 11: 3 Material and Methods. “…for 2 h at room temperature overnight.” 2 hours or overnight, not both.
Response: “overnight” is correct; therefore, we deleted “2 h.”
Comment 12: “..1,200 collagen fibrils from four TEM images of two mice..” Was the mean value of each mice used for the statistics? N=2? If not, a t-test is not appropriate.
Response: We performed electron micrographs of the tendons of three control and Cant1–/– mice. We have corrected as follows: “Values represent means ± SD (n = 3). We calculated the mean value of fibrils (approximately 500) in the pictures for one mouse and compared them for the Ctrl and Cant1–/– mice.” Fig. 3B and the corresponding legend have been revised accordingly.
Comment 13: 4. Tensile testing. Please provide more information on the test apparatus. Which measurement range/sensitivity had the load cell?
Response: We have added this information to 4.4. section of the main text.
Comment 14: Please provide the ARRIVE documentation.
Response: We have submitted the ARRIVE documentation.
Reviewer 2 Report
Comments and Suggestions for Authors
This paper investigates the role of Calcium-Activated Nucleotidase 1 (CANT1) in tendon collagen fibrogenesis, focusing on its impact on dermatan sulfate (DS) and chondroitin sulfate (CS) synthesis and decorin function. The study uses Cant1 knockout mice to demonstrate that CANT1 dysfunction leads to defective DS/CS synthesis, impaired decorin function, and ultimately, abnormal collagen fibril formation. While the study demonstrates a correlation between CANT1 dysfunction and impaired collagen fibrogenesis, the precise molecular mechanisms underlying this relationship remain unclear. The authors should consider performing additional experiments to investigate the molecular mechanisms by which CANT1 regulates DS/CS synthesis and decorin function. This could involve examining the expression of genes involved in GAG synthesis or analyzing the post-translational modifications of decorin.
Introduction:
- The introduction mentions the role of nucleotide sugar transporters (NSTs) in GAG synthesis. Could CANT1 deficiency affect the function or expression of NSTs, and could this contribute to the observed reduction in DS/CS content?
- The authors hypothesize that impaired GAG biosynthesis and subsequent SLRP dysfunction are responsible for tendon fragility in DBQD1 patients. Are there any other potential mechanisms by which CANT1 dysfunction could lead to tendon fragility, and were these considered in the study?
Methods:
- For the tensile tests, how was the gauge length of the collagen fascicles determined, and how might variations in gauge length affect the results?
- In the disaccharide composition analysis, what steps were taken to ensure that the GAGs were completely depolymerized and that the disaccharides were accurately quantified?
Results:
- Regarding Figure 2F, the maximum load of the Cant1-/- tail tendons was significantly decreased, but the tensile strength, strain at failure, and tangent modulus were not significantly different. Can you explain in more detail why the decrease in maximum load doesn't affect other measures of elasticity?
- In Figure 3, the collagen fibrils in Cant1−/− tendons appear more tightly packed. Was there any quantification done to assess collagen packing density and how it might affect the mechanical properties of the tendon?
- Table 1 shows a reduction in CS and DS content in Cant1-/- tendons. Were there any changes observed in the sulfation patterns of these GAGs, and could these changes contribute to the observed phenotypes?
- The study mentions mild wavy tails in Cant1−/− Were there any other skeletal abnormalities observed in these mice that might be relevant to the tendon phenotype?
- Were there any changes in the expression levels of other ECM components (e.g., collagen types, other proteoglycans) in Cant1−/− tendons, and could these changes contribute to the observed phenotypes?
Discussion:
- The discussion focuses on the role of decorin in collagen fibrogenesis. Could CANT1 deficiency affect the interaction between decorin and other ECM components, such as collagen fibrils or other proteoglycans, and could this contribute to the observed phenotypes?
- The authors suggest that multiple dislocations in DBQD1 patients may be attributed to tendon fragility resulting from CANT1 dysfunction. Are there any other factors that could contribute to multiple dislocations in DBQD1 patients, and how might these factors interact with tendon fragility?
Minor revisions:
- Abstract, Line 6: Change "Decorin is a proteoglycan, in which one dermatan sulfate (DS) or chondroitin sulfate (CS) chain, a type of GAG, is attached to its core protein..." to "Decorin is a proteoglycan with one dermatan sulfate (DS) or chondroitin sulfate (CS) chain (a type of GAG) attached to its core protein..."
- Introduction, Line 15: Change "These PGs, which belong to the small leucine-rich proteoglycan (SLRP) family containing numerous leucine-rich repeats, are important regulators of ECM assembly and cell signaling in the connective tissue." to "These PGs, belonging to the small leucine-rich proteoglycan (SLRP) family, which contain numerous leucine-rich repeats, are important regulators of ECM assembly and cell signaling in connective tissue."
- Results, Section 2.1, Paragraph 2: "After infancy, Cant1−/− mice exhibited mild wavy tails (Figure S1)." should be changed to "Cant1−/− mice exhibited mild wavy tails after infancy (Figure S1)."
- Figure 2 Caption: The phrase "The wild-type and Cant1+/− mice were used as control mice." is repeated in Figures 1 and 2. It may be better to indicate this in the "Materials and Methods" section to avoid repetition.
- Throughout the manuscript: Consider standardizing the abbreviation of control as "Ctrl" (as used in some places) or "control" (written out) for consistency.
Author Response
Comment 1: While the study demonstrates a correlation between CANT1 dysfunction and impaired collagen fibrogenesis, the precise molecular mechanisms underlying this relationship remain unclear. The authors should consider performing additional experiments to investigate the molecular mechanisms by which CANT1 regulates DS/CS synthesis and decorin function. This could involve examining the expression of genes involved in GAG synthesis or analyzing the post-translational modifications of decorin.
Response: We believe that the CANT1 dysfunction shortens the DS chain attached to decorin, leading to impaired decorin function in the assembly of collagen fibrils in tendons. As Reviewer 2 pointed out, the exact molecular mechanism has not been elucidated, but the importance of the integrity of DS chains attached to decorin in the assembly of collagen fibrils has been demonstrated in several studies on musculocontractural Ehlers-Danlos syndrome caused by CHST14 mutations, as discussed in paragraph 5 of the Discussion section. Genes involved in GAG synthesis and the post-translational modifications of decorin may play a role in the pathogenesis of Cant1–/– tendons. However, we believe that they do not play a major role in its pathogenesis and the defective GAG synthesis does. Because the required analyses are beyond the scope of additional experiments, I would be happy to let them work in the future.
(Introduction)
Comment 2: The introduction mentions the role of nucleotide sugar transporters (NSTs) in GAG synthesis. Could CANT1 deficiency affect the function or expression of NSTs, and could this contribute to the observed reduction in DS/CS content?
Response: We have added the following text to paragraph 2 of the Discussion section (additions are indicated in red): “CANT1 preferentially hydrolyzes UDP to UMP and phosphate in the ER and Golgi apparatus. The generated UMPs are used as substrates for the antiport mediated by the NSTs. Therefore, the reduced UMP production due to CANT1 deficiency should reduce the NST activity. As a result, there are limited nucleotide sugars in the ER/Golgi lumen, leading to a reduced DS/CS content.”
Comment 3: The authors hypothesize that impaired GAG biosynthesis and subsequent SLRP dysfunction are responsible for tendon fragility in DBQD1 patients. Are there any other potential mechanisms by which CANT1 dysfunction could lead to tendon fragility, and were these considered in the study?
Response: Other reported functions of CANT1, beyond GAG synthesis, are stated in paragraph 8 of the Discussion section. However, the relationship between the defects in these CANT1 functions beyond GAG synthesis and tendon fragility remains unclear.
(Methods)
Comment 4: For the tensile tests, how was the gauge length of the collagen fascicles determined, and how might variations in gauge length affect the results?
Response: We have added this information to Section 4.4. of the main text.
Comment 5: In the disaccharide composition analysis, what steps were taken to ensure that the GAGs were completely depolymerized and that the disaccharides were accurately quantified?
Response: Because the excess amounts of GAG-degrading enzymes were utilized for the depolymerization of each GAG, they were supposed to be digested completely. Each chromatogram of the digest is shown in Figure S3. Furthermore, to ensure the enzymatic activity, commercial chondroitin sulfate-A, heparin, heparan sulfate, and keratan sulfate-I were also treated with the respective GAG-degrading enzymes.
(Results)
Comment 6: Regarding Figure 2F, the maximum load of the Cant1–/– tail tendons was significantly decreased, but the tensile strength, strain at failure, and tangent modulus were not significantly different. Can you explain in more detail why the decrease in maximum load doesn't affect other measures of elasticity?
Response: Significant reductions in the maximum load but no changes in the tensile strength, strain at failure, and tangential coefficient have been reported in several mutant mouse analyses, including Mkx–/– and tendon-specific Rcn3–/– mice. And 1,2). The fact that there was no change in the tensile strength and strain at failure indicates that the material property was not affected. Mkx is a transcription factor that is important for tenocyte differentiation, and Mkx–/– mice show more severe tendon hypoplasia than Cant1–/– mice. It should be quite difficult to detect the changes in the material property of mouse tendons in current tensile tests, because even Mkx–/– mice showing such severe tendon hypoplasia do not show any change.
1) The Mohawk homeobox gene is a critical regulator of tendon differentiation. Proc Natl Acad Sci USA 107(23): 10538–42, 2010.
2) Reticulocalbin 3 is involved in postnatal tendon development by regulating collagen fibrillogenesis and cellular maturation. Sci Rep 11: 108686, 2021.
Comment 7: In Figure 3, the collagen fibrils in Cant1−/− tendons appear more tightly packed. Was there any quantification done to assess collagen packing density and how it might affect the mechanical properties of the tendon?
Response: We quantified the fibril density in the images and added the corresponding figure (Figure 3C). The density of Cant1−/− tendons was significantly higher than that of Ctrl tendons, indicating that the collagen fibrils in Cant1−/− tendons were more tightly packed than those in Ctrl tendons. However, the relationship between the collagen fibril density and mechanical properties remains unclear.
Comment 8: Table 1 shows a reduction in CS and DS content in Cant1−/− tendons. Were there any changes observed in the sulfation patterns of these GAGs, and could these changes contribute to the observed phenotypes?
Response : As shown in Supplementary Tables S1-S3, the amounts of each disaccharide in the Cant1–/– tendons were significantly reduced compared to those in the Ctrl tendons. However, no obvious changes were observed in the sulfation pattern (ratio of the respective disaccharides). Therefore, the phenotype of Cant1–/– mice should appear to be dependent on the reduction in CS/SD content, but not on the changes in sulfation patterns. “The sulfation patterns in the CS and DS were comparable between Ctrl and Cant1−/− mice (Tables S1–S3).” was added to Section 2.3 of the main text.
Comment 9: The study mentions mild wavy tails in Cant1–/– mice, were there any other skeletal abnormalities observed in these mice that might be relevant to the tendon phenotype? Were there any changes in the expression levels of other ECM components (e.g., collagen types, other proteoglycans) in Cant1–/– tendons, and could these changes contribute to the observed phenotypes?
Response: Because the GAG content is reduced in the cartilage of Cant1–/– mice, the cartilage defects due to the reduced GAG contents may be associated with the wavy tail phenotype. Eighty % of proteoglycans (PGs) present in the tendon are decorin, and the content of other PGs is low. We performed western blot analyses to detect other PGs, but could not detect them. We observed no change in the mRNA expression of type I collagen (Col1a1), and we did not examine other collagens. In the future, we would like to address this issue.
(Discussion)
Comment 10: The discussion focuses on the role of decorin in collagen fibrogenesis. Could CANT1 deficiency affect the interaction between decorin and other ECM components, such as collagen fibrils or other proteoglycans, and could this contribute to the observed phenotypes?
Response: “The tendons contain other SLRPs, including biglycan, lumican, and fibromodulin, which are also involved in tendon fibrogenesis, suggesting that the CANT1 deficiency also affects the GAG side chains of these SLRPs. The defective functions of these SLRPs may affect the pathogenesis of Cant1−/− tendons.” has been added in paragraph 3 of the Discussion section. Eighty % of the PGs present in the tendons are decorin, and the content of other PGs is low. We performed western blot analyses to detect other PGs, but could not detect them. Therefore, it is difficult to discuss the detailed relationship between these SLRPs and CANT1 functions in the tenons.
Comment 11: The authors suggest that multiple dislocations in DBQD1 patients may be attributed to tendon fragility resulting from CANT1 dysfunction. Are there any other factors that could contribute to multiple dislocations in DBQD1 patients, and how might these factors interact with tendon fragility?
Response: The most common cause of hereditary joint dislocations, including genetic skeletal disorders, classified in Group 5, is thought to be the tendon fragility. Many skeletal disorders classified in Group 5 involve abnormal GAG production, indicating that GAG abnormalities are the cause of joint dislocation. GAGs are also abundant in articular cartilage; however, there are no reports indicating a relationship between abnormal GAG metabolism in articular cartilage and joint dislocation. In rheumatoid arthritis, the production of synovial fluid is reduced, resulting in joint swelling and stiffness, which may dislocate when a strong force is applied. CANT1 dysfunction does not cause joint inflammation, thereby negating this theory.
(Minor revisions)
Comment 12: Abstract, Line 6: Change "Decorin is a proteoglycan, in which one dermatan sulfate (DS) or chondroitin sulfate (CS) chain, a type of GAG, is attached to its core protein..." to "Decorin is a proteoglycan with one dermatan sulfate (DS) or chondroitin sulfate (CS) chain (a type of GAG) attached to its core protein..."
Response: We have corrected the main text, as instructed.
Comment 13: Introduction, Line 15: Change "These PGs, which belong to the small leucine-rich proteoglycan (SLRP) family containing numerous leucine-rich repeats, are important regulators of ECM assembly and cell signaling in the connective tissue." to "These PGs, belonging to the small leucine-rich proteoglycan (SLRP) family, which contain numerous leucine-rich repeats, are important regulators of ECM assembly and cell signaling in connective tissue."
Response: We have corrected the main text, as instructed.
Comment 14: Results, Section 2.1, Paragraph 2: "After infancy, Cant1−/− mice exhibited mild wavy tails (Figure S1)." should be changed to " Cant1−/− mice exhibited mild wavy tails after infancy (Figure S1)."
Response: We have corrected the main text, as instructed.
Comment 15: Figure 2 Caption: The phrase "The wild-type and Cant1−/− mice were used as control mice." is repeated in Figures 1 and 2. It may be better to indicate this in the "Materials and Methods" section to avoid repetition.
Response: We have added the following sentence in Section 4.1 of the main text: Given that the Cant1+/− mice did not present an overt phenotype, the wild-type and Cant1+/− littermates were used as Ctrl mice. We have deleted this information from the legends of Figures 1 and 2.
Comment 16: Throughout the manuscript: Consider standardizing the abbreviation of control as "Ctrl" (as used in some places) or "control" (written out) for consistency.
Response: We have unified this by using the abbreviation "Ctrl" for the control.
Reviewer 3 Report
Comments and Suggestions for Authors
This is a phenotype study of CANT1 gene knockout mice to explore the potential role of CANT1 in tendon collagen production. Unfortunately, there is a lack of pathological models to further explore the regulatory mechanisms of CANT1 in tendon related diseases.
Comments on the Quality of English LanguageIt is recommended for native English speakers to revise this manuscript.
Author Response
Comment 1: This is a phenotype study of CANT1 gene knockout mice to explore the potential role of CANT1 in tendon collagen production. Unfortunately, there is a lack of pathological models to further explore the regulatory mechanisms of CANT1 in tendon related diseases.
Response: We agree that there is a lack of pathological models to further explore the regulatory mechanisms of CANT1 in tendon related diseases. However, we think that these analyses are beyond the scope of additional experiments. We would like this to be a topic for future research.
Comment 2: It is recommended for native English speakers to revise this manuscript.
Response: We have edited our manuscript using MDPI author services and submitted the English proofreading certificate.
Round 2
Reviewer 1 Report
Comments and Suggestions for Authors
The authors have taken the comments and suggestions into account and modified the manuscript accordingly.
Reviewer 2 Report
Comments and Suggestions for Authors
The revisions have adequately addressed the previously identified deficiencies. The manuscript now appears to have reached a level of quality and completeness that makes it suitable for acceptance.